# A Novel Formulation of Asparaginase Encapsulated into Virus-like Particles of Brome Mosaic Virus: In Vitro and In Vivo Evidence

**DOI:** 10.3390/pharmaceutics15092260

**Published:** 2023-08-31

**Authors:** Francisca Villanueva-Flores, Ana Ruth Pastor, Laura A. Palomares, Alejandro Huerta-Saquero

**Affiliations:** 1Departamento de Bionanotecnología, Centro de Nanociencias y Nanotecnología, Universidad Nacional Autónoma de México, Km. 107 Carretera Tijuana-Ensenada, Ensenada 22860, BC, Mexico; 2Departamento de Medicina Molecular y Bioprocesos, Instituto de Biotecnología, Universidad Nacional Autónoma de México, Ave. Universidad 2001, Col. Chamilpa, Cuernavaca 62210, MO, Mexico; ruth.pastor@ibt.unam.mx (A.R.P.); laura.palomares@ibt.unam.mx (L.A.P.); 3Tecnológico de Monterrey, Escuela Nacional de Medicina y Ciencias de la Salud, Avenida Heroico Colegio Militar 4700, Nombre de Dios, Chihuahua 31300, CH, Mexico

**Keywords:** brome mosaic virus, asparaginase, acute lymphoblastic leukemia, encapsulation, virus-like particles

## Abstract

The interest in plant-derived virus-like particles (pVLPs) for the design of a new generation of nanocarriers is based on their lack of infection for humans, their immunostimulatory properties to fight cancer cells, and their capability to contain and release cargo molecules. Asparaginase (ASNase) is an FDA-approved drug to treat acute lymphoblastic leukemia (LLA); however, it exhibits high immunogenicity which often leads to discontinuation of treatment. In previous work, we encapsulated ASNase into bacteriophage P22-based VLPs through genetic-directed design to form the ASNase-P22 nanobioreactors. In this work, a commercial ASNase was encapsulated into brome mosaic virus-like particles (BMV-VLPs) to form stable ASNase-BMV nanobioreactors. According to our results, we observed that ASNase-BMV nanobioreactors had similar cytotoxicity against MOLT-4 and Reh cells as the commercial drug. In vivo assays showed a higher specific anti-ASNase IgG response in BALB/c mice immunized with ASNase encapsulated into BMV-VLPs compared with those immunized with free ASNase. Nevertheless, we also detected a high and specific IgG response against BMV capsids on both ASNase-filled capsids (ASNase-BMV) and empty BMV capsids. Despite the fact that our in vivo studies showed that the BMV-VLPs stimulate the immune response either empty or with cargo proteins, the specific cytotoxicity against leukemic cells allows us to propose ASNase-BMV as a potential novel formulation for LLA treatment where in vitro and in vivo evidence of functionality is provided.

## 1. Introduction

There is a growing interest in plant-derived virus-like particles (pVLPs) for establishing novel platforms inspired by nature to develop novel nanobiomaterials for biomedical applications. pVLPs have tremendous potential for biomedical applications because: (1) plant viruses are noninfectious for humans; (2) they have a tunable surface; (3) pVLPs exhibit immunostimulatory properties to fight cancer cells; (4) they are stable, biocompatible, and biodegradable; (5) they do not contain pathogens hazardous to mammals; (6) unlike inorganic nanocarriers, pVLPs do not accumulate in the body at toxic levels; (7) plant virus proteins are capable of forming large assemblies and layers, with putative applications for tissue engineering; (8) pVLPs can be produced at low cost and their manufacturing is highly scalable; (9) pVLPs can be self-assembled and spontaneously wrap cargo molecules inside their shell with a high load capacity [1,2,3].

pVLPs are promising biomedical tools, and currently, some of them are close to being commercialized. Applications of pVLPs include the development of novel nanocarriers for vaccines against cancer and infectious diseases, agents in cancer immunotherapy, nanocarriers for gene and drug delivery, single- and multi-layered scaffolds, and biosensors, among others [4,5,6,7].

The most common pVLPs found in the literature are tobacco mosaic virus (TMV) and potato virus X (PVX). Both viruses have rod-shaped structures with high aspect ratios, which make them ideal platforms for drug and gene delivery [8,9,10]. On the other hand, pVLPs based on cowpea chlorotic mottle virus (CCMV) possess an icosahedral structure with unique benefits as a nanoparticle, such as epitope display, drug and gene delivery, and anti-cancer properties. This is relevant because it has been reported that different shapes of pVLPs trigger different responses in vivo [11,12,13].

In contrast, brome mosaic virus (BMV) has been mostly used as a model for studying virus–host interactions, gene expression, RNA replication, and recently, siRNA encapsulation [14,15,16,17]. BMV is an isometric, nonenveloped, positive-strand RNA virus, which belongs to the Bromovirus genus, and the *Bromoviridae* family, in the alphavirus-like superfamily. BMV can infect many species of Gramineae and dicotyledonous plants. BMV is widely distributed around the world, and it is not considered an important plague that causes economic losses in cereals crops. BMV can be transmitted by sap inoculation through mechanical damage, aphids, wheat stem rust, seed, and *Xiphinema* and *Coleoptera* species. BMV infection induces a mosaic or streaked pattern of chlorosis on infected leaves and growth retardation [14,18,19].

The BMV capsid is 28 nm in diameter with an inner core diameter of around 18 nm. The outer capsid is composed of 180 copies of a 20 kDa capsid protein (CP), arranged in a *T* = 3 lattice (Caspar-Klug’s triangulation numbers), and it shows an icosahedral symmetry [20].

The BMV has tremendous potential for biotechnological applications; however, it has been poorly exploited. Diverse strategies for encapsulating and releasing molecules into/from BMV-VLPs through the understanding of the electrostatic interactions in the assembly and disassembly of capsid proteins have been explored. Electrostatic interactions between BMV proteins can be easily modulated by varying the pH and ionic strength. The *N*-terminal of the CP is composed of 26 amino acids, and it is rich in basic residues, such as arginine and lysine, and exhibits a high pK_a_. This feature is shared by other icosahedral plant viruses [21,22,23,24]. The *N*-terminus is inside the capsid and remains positively charged over a wide range of pH. Thereby, spontaneous packaging into the BMV capsid is favored by an interaction between virus proteins and negatively charged molecules. The isoelectric point (pI) of native BMV is 5.2, and the pI of BMV CP is 6.5. At the pI, electrical charges are fully neutralized, and hydrophobic interactions dominate and stabilize the capsid. This property is advantageous considering that an acidic pH is typical of cancerous microenvironments because of anaerobic glycolysis [25], allowing the use of enzymatic nanoreactors for cancer treatment. According to Duran-Meza et al. [26], this knowledge can be exploited to encapsulate negatively charged cargo molecules inside the BMV capsid [27,28,29,30]. Recently, Gama et al. [31] developed the first nanobioreactor based on BMV-VLPs by the encapsulation of galactose-1-phosphate uridylyl-transferase (GALT). They demonstrated that the nanobioreactors were internalized into different mammalian cell lines and they proposed them as an alternative treatment for classic galactosemia.

In this work, we encapsulated asparaginase from *E. coli* (EC 3.5.1.1) (ASNase) into BMV-VLPs to form the nanobioreactor called ASNase-BMV. ASNase is a homotetrameric enzyme that catalyzes the hydrolysis of asparagine (Asn) to form aspartate (Asp) and ammonium. This enzyme has been widely used for the treatment of acute lymphoblastic leukemia (ALL) in children. ASNase reduces the Asn levels in the blood, leading to starvation of the leukemic cells, which die by apoptosis [32,33]. However, one of the major drawbacks of ASNase treatment is its high immunogenicity. High ASNase immunogenicity often leads to discontinuation of treatment or, to the prescription of alternative but considerably more expensive ASNase formulations. Based on this, ASNase encapsulation into BMV-VLPs is an interesting proposal to reduce, or at least delay, immunogenic responses [34,35]. Interestingly, Nuñez-Rivera et al., [17] have demonstrated that empty BMV-VLPs do not activate macrophages in vitro, which suggests that BMV-VLPs have low immunogenicity, being a nonimmunogenic nanocarrier.

Overall, this work aims to encapsulate ASNase into BMV-VLPs, to characterize its enzymatic profile, and to evaluate the cytotoxicity of ASNase-BMV against two human leukemia cell lines. In addition, we immunized female BALB/c mice with empty BMV-VLPs, ASNase encapsulated into BMV-VLPs, free ASNase, and a formulation buffer (as a mock group), and we analyzed sera for specific IgG profiles against BMV capsids and ASNase. To our knowledge, this is the first complete study showing physicochemical characterization and in vivo assays of ASNase encapsulated into BMV-VLPs.

## 2. Materials and Methods

### 2.1. Propagation of BMV Virions

Barley seeds (*Hordeum vulgare*) were washed with 0.5% sodium hypochlorite (Cloralex^®^, Nuevo León, Mexico) and abundant distilled water. Seeds were soaked overnight in distilled water. Then, the water was decanted, and the seeds were germinated in a glass jar tilted 45° on a wet cheesecloth fabric for three days, or until radicles reached 3 to 5 mm in length. Germinated seeds were transplanted to potting soil in a greenhouse with a natural 12/12 light and dark photoperiod. Plants were drip watered for 2–3 weeks, or until the first leaves reached 10 cm high. Plants were infected by a smooth leaf abrasion with virgin polypropylene fibers (Concret^®^, Mexico City, Mexico). Infection was performed with 10 µL of inoculum containing 200 µg/mL of wildtype BMV dispersed in 10 mM sodium phosphate buffer (Na_2_HPO_4_, Merck S9763, and NaH_2_PO_4_, Merck S3139) at pH 6, enriched with 10 mM MgCl_2_ (Merck, M9272). After 2 to 3 weeks, chlorotic leaves were collected and frozen at −80 °C until use. Plants were able to continue producing chlorotic leaves for 4 to 6 additional weeks.

### 2.2. BMV Virions Purification

All solutions used for BMV purification were previously filtered through nylon filter membranes of 0.22 µm pore size (Merck, Rahway, NJ, USA, Z290807) and sterilized by autoclaving. Frozen leaves were ground in a blender (Oster^®^, Neosho, MI, USA) with 2 mL of cold extraction buffer for each gram of vegetal tissue used. The extraction buffer contained 500 mM sodium acetate (Merck, 791741) and 80 mM magnesium acetate (Merck, 228648), pH 4.5. Afterwards, the solution was filtered using a cheesecloth fabric, then an identical volume of chloroform (Merck, 650498) was added. The mixture was gently shaken for 5 min and centrifuged at 10,000 rpm for 40 min at 4 °C (Avanti JXN Beckman, rotor JA-14, Brea, CA, USA). The supernatant (aqueous, yellow phase) was recovered and centrifuged at 10,000 rpm for 25 min at 4 °C (Avanti JXN Beckman, rotor JA-14). The supernatant was recovered and stirred at 4 °C for 3 h or overnight. The supernatant was placed on a 10% sucrose (Merck Millipore, Burlington, MA, USA, 1076870250) cushion to separate the protein fraction in a ratio of 5:1 *v/v* (supernatant and sucrose cushion, respectively). Samples were ultracentrifuged at 32,000 rpm for 2 h at 4 °C using a Beckman Optima XPN-100 ultracentrifuge, rotor SW 32 Ti. The supernatant was stored at 4 °C until use. Pellets were resuspended with 500 µL of viral suspension buffer (50 mM sodium acetate, Merck 79174, and 8-mM magnesium acetate, Merck 228648, pH 4.5). Suspensions were pooled and centrifuged at 5000 rpm for 10 min at 4 °C using a microcentrifuge (Eppendorf^TM^, Hamburg, Germany, model 5418).

Continuous sucrose gradients were prepared by 4 freeze (−80 °C)/thaw (room temperature) cycles of a 25% *w*/*v* sucrose solution in viral suspension buffer. The pooled suspension was gently added above the sucrose gradients, avoiding disturbing them. Sucrose gradients were ultracentrifuged at 30,000 rpm for 2 h at 4 °C (Beckman Optima XPN-100 ultracentrifuge, rotor SW 32 Ti). In a dark room, tubes were illuminated with white light. A brilliant blue band was observed in half of the ultracentrifuge tube. The blue fraction was collected with a glass Pasteur pipette. The blue fraction was diluted in a ratio of 1:4 *v*/*v* with the viral suspension buffer and ultracentrifuged at 32,000 rpm for 3 h at 4 °C (Beckman Optima XPN-100 ultracentrifuge, rotor SW 32 Ti). The pellet was resuspended with the same buffer and quantified as described in the next section.

### 2.3. BMV Virions Quantification

Virus concentration was calculated by measuring the UV absorbance at 260 nm (A_260nm_) using a Nanodrop 2000c spectrophotometer (Thermo Fischer Scientific, Waltham, MA, USA), according to Equation (1).
(1)$$Virus\;concentration\;\left(\frac{\mathrm{mg}}{\mathrm{mL}}\right)=\frac{A_{260\mathrm{nm}}}{5.8}\;$$
where 5.8 corresponds to the virus molar extinction coefficient [36].

The ratio $\frac{A_{260\mathrm{nm}}}{A_{280\mathrm{nm}}}\geq 1.5$ was considered as a quality criterion for purity samples.

### 2.4. ASNase Encapsulation into BMV-VLPs

Commercial ASNase (Leunase^®^, Sanfer, Akita, Japan) was encapsulated into BMV-VLPs using a Leunase^®^/CP molar ratio of 0.07 to form ASNase-BMV nanobioreactors. Encapsulation was performed by taking advantage of the electrostatic interactions between Leunase and capsid proteins, following the procedure previously reported by Gama et al. [31]. To remove the nonencapsulated enzyme, ASNase-BMV nanobioreactors were ultracentrifuged at 32,000 rpm for 2 h at 4 °C on a 10% sucrose cushion. Pellets were resuspended with 500 µL of PBS and then dialyzed with the same buffer at 4 °C for 24 h using Spectra/Por^®^ 1 MWCO 6000–8000 membranes (Spectrum^®^ Laboratories, Washington, DC, USA).

### 2.5. Characterization of ASNase-BMV Nanobioreactors

#### 2.5.1. Dynamic Light Scattering (DLS) Characterization

Sizes of BMV-VLPs and ASNase-BMV were determined by dynamic light scattering (DLS, Zetasizer NanoZS, Malvern, UK). The samples were dispersed in 50 mM PBS, pH 7, at a concentration of 10 μg/mL. Analysis was performed using Malvern DTS1070 Zetasizer folded capillary cells.

#### 2.5.2. Transmission Electron Microscopy (TEM) Characterization

The size and shape of BMV-VLPs and ASNase-BMV were determined by TEM. First, 2 µg of BMV-VLPs and ASNase-BMV were placed on a copper grid (400 mesh, formvar/carbon, TedPella, Redding, CA, USA) and incubated for 2 min, then the excess was removed with a Whatman filter paper (Merck, WHA1001325). The samples were stained with 6 μL of 2% uranyl acetate (Fisher Scientific, NC0788109) for 1 min. The excess stain was removed by capillarity with filter paper. Samples were analyzed with a JEOL JEM-2010 TEM operated at 200 kV.

#### 2.5.3. Determination of Enzymatic Activity

Asparaginase enzymatic activity was performed using the principle of the Berthelot’s reaction [37]. Asparagine deamidation by asparaginase at different reaction times was performed. Ammonium was quantified according to the procedure previously reported by Díaz-Barriga et al. [38].

#### 2.5.4. Thermal Stability

Thermal stability of free and encapsidated (ASNase-BMV) asparaginases was studied by measuring the enzymatic activity at 25° C and 37 °C for at least 96 h. Enzymatic activity was determined in the presence of 10% *v*/*v* of glycerol (Merck, G9012) as a stabilizer.

### 2.6. Cytotoxicity Activity of ASNase-BMV

#### 2.6.1. Cell Culture and Cytotoxicity Assays

The T-cell lymphoblastic human leukemia cell line, MOLT-4 (CRL-1582, ATCC), and the human B-cell line, Reh (CRL-8286, ATCC), were cultured in the Roswell Park Memorial Institute medium (RPMI 1640) (Merck, R6504), complemented with 2 mM L-glutamine (Merck, G6392), 10 mM HEPES (Sigma, St. Louis, MO, USA, H4034), 1 mM sodium pyruvate (Merck, P5280), 4500 mg/L glucose (Merck, G7021), 1500 mg/L sodium bicarbonate (Merck, S5761), and 10% fetal bovine serum (*v*/*v*) (Gibco, 11533387). Cells were incubated at 37 °C in 5% CO_2_. MOLT-4 and Reh cells were plated on 96 well-plates (Thermo Fisher Scientific, 260860) at 4 × 10^4^ cells/mL, and exposed to 1 IU/mL of encapsulated or nonencapsulated ASNase for 132 h. Cell proliferation was measured using the CellTiter 96^®^ Aqueous One Solution Cell Proliferation Assay kit (Promega, Madison, WI, USA, G3581), according to the manufacturer’s instructions. Absorbance was measured in a Multiskan FC, Thermo Fisher Scientific, 51119000 spectrometer. Viability of treated wells was normalized to that of untreated wells, using Equation (2).
(2)$$Normalized\;cell\;viability=\frac{A_{Treated\;well}}{A_{control\;well}}\times 100$$
where *A* is the absorbance at 490 nm.

#### 2.6.2. Mice Immunization

Female BALB/c mice between six- to eight-week-old were used for all immunization studies. No additives or adjuvants were used. Groups of five mice were immunized intraperitoneally with different formulations of BMV-VLPs. Group A was immunized with 90 µg of empty BMV-VLPs; group B was immunized with 90 µg of ASN encapsulated into BMV-VLPs (containing around 300 µg of BMV-VLPs protein); group C, as a control group, was immunized with the formulation buffer; and group D was immunized with 90 µg of commercial asparaginase (Leunase). Three doses were applied as follows: the first on day 0, the second on day 14, and the third on day 28. Pre-immune sera were collected seven days before the first immunization. To measure the anti-BMV-VLPs and anti-ASN humoral response, individual blood samples were collected through the tail vein on days 7, 21, 35, and 42 after the first immunization. The IBt bioethics committee on the use and care of animals approved all experiments.

#### 2.6.3. IgG immune Responses

Anti-BMV IgG antibodies were measured in mice sera samples by ELISA. Briefly, 100 µL of a solution of purified BMV-VLP at 5 µg/mL was adsorbed onto 96-well microplates for at least 16 h at 2–8 °C. Following the incubation time, plates were washed 3 times with Tris buffer to remove unbound BMV-VLP, blocked with 2 mg/mL of gelatin in Tris buffer, pH 8, to prevent nonspecific binding, and incubated for 2 h at 37 °C. Then, serial dilutions of mice sera at 100 µL of final volume were prepared and loaded into the plate. After 2 h of incubation at 37 °C, plates were washed 5 times, and an alkaline phosphatase-conjugated goat anti-mouse IgG antibody (Jackson ImmunoResearch, West Grove, PA, USA) in a 1:1500 dilution was added. Plates were incubated for 1 h at 37 °C. Next, plates were washed again 5 times, 100 µL of phosphatase substrate at 120 µg/mL (Sigma, P4744) was added and read at 405 nm in a microplate reader (FLUOstar Omega, BMG Labtech, Ortenberg, Germany). IgG titers were calculated as the reciprocal of the serum dilution at which the response was 50%.

Anti-ASNase IgG antibodies were measured with an in-house ELISA assay. Briefly, 100 µL of a solution of ASNase at 5 µg/mL was adsorbed into 96-well plates. After at least 16 h of incubation at 2–8 °C, plates were washed and blocked for 2 h at 37 °C with a 2 mg/mL gelatin solution. Next, 1:3 serial dilutions of mice sera were prepared and added to plates and incubated for 2 h at 37 °C, then plates were washed, and an alkaline phosphatase-conjugated goat anti-mouse IgG antibody (Jackson ImmunoResearch, West Grove, PA, USA) in a 1:1500 dilution was added. After 1 h, 100 µL of phosphatase substrate at 120 µg/mL (Sigma, P4744) was added to the plates and read at 405 nm in a microplate reader. Titers were calculated as the reciprocal of the serum dilution at which the response was 50%.

#### 2.6.4. Statistical Analysis

All measurements reported are displayed as the average ± standard deviation of at least three independent experiments. Statistical significances were calculated by a two-way analysis of variance (ANOVA) followed by a Tukey’s test.

For all results in mice, the average ± standard deviation of five mice in each group is shown. Statistical significances were calculated by Student’s *t*-test using Sigma Plot software.

## 3. Results and Discussion

### 3.1. BMV-VLPs and ASNase-BMV Characterization

#### Size and Shape of Nanobioreactors

BMV virions were obtained by infection of barley leaves, then virions were purified, disassembled, and reassembled to encapsulate commercial ASNase, according to the protocol previously reported by Gama et al. [31]. After ASNase encapsulation, purification of nanoreactors is essential to characterize biological and enzymatic properties. By ultracentrifugation on the sucrose cushion, the nonencapsulated enzyme remains in the supernatant, whereas encapsulated asparaginase inside BMV capsids precipitates, allowing their purification. The size and aggregation state of BMV-VLPs and ASNase-BMV were determined by TEM and DLS. Results are shown in Figure 1.

Figure 1a shows that the size of BMV-VLPs was 30 ± 7.9 nm in diameter, which agrees with previous reports [20]. Figure 1b shows that ASNase-BMV nanobioreactors are 31.2 ± 8.4 nm in diameter, which means that ASNase encapsulation did not affect the particle size. Both samples showed minoritarian aggregates around 622 and 1424 nm in size, respectively. The morphology of nanoparticles was evaluated by TEM. Micrographs of empty BMV-VLPs and ASNase-BMV nanobioreactors are shown in Figure 1c and Figure 1d, respectively. As can be observed in the images, both BMV-VLPs and ASNase-BMV have the expected icosahedral morphology previously reported by Lucas et al. [20]. The observed size correlates to the DLS results. It is worth highlighting that approximately 50% of BMV-VLPs show a dense black inner cavity, corresponding to empty capsids. So that we can infer that nearly 50% of the encapsulation efficiency of ASNase was achieved.

Our results show that we obtained nanoparticles with the expected size and shape, and that the asparaginase encapsulation was carried out with approximately 50% efficiency.

### 3.2. Asparaginase Enzymatic Characterization

Characterizing the enzymatic profile of nanobioreactors is critical to determine a priori their potential as a drug formulation. As has been pointed out by multiple works, the catalytic performance of an enzyme can vary depending on its encapsulation conditions, such as confinement density, substrate diffusion, or molecular dynamics limitations inside the VLPs. This behavior is independent of the nature of the nanocontainer used, so a decreased enzymatic performance in encapsulated enzymes is a common observation in the literature [35,39,40,41].

In previous work, we encapsulated ASNase into capsids of the bacteriophage P22. The Michaelis–Menten constant (K_M_) of ASNase was 15-fold higher when encapsulated, which indicates that the affinity for the substrate drops, due to a limited substrate diffusion across the capsid, or to a reduced enzyme movement inside the capsid [38].

In this work, we followed a different encapsulation strategy, using BMV capsids as a nanocontainer. We hypothesized that an encapsulation based on electrostatic interactions instead of a covalent attachment inside the capsid could favor the enzymatic performance. In Figure 2, the enzymatic kinetics of free ASNase (Figure 2a) and ASNase-BMV (Figure 2b) are shown. Berthelot’s reaction was used to determine the ammonia produced as a product of the asparaginase reaction. Assays were performed by varying the asparagine (Asn) concentration from 0.5 mM to 6 mM, resulting in the typical hyperbolic curves. According to our results, the initial velocity of ammonia production (V_0_) increased with the substrate concentration. The graphs of Asn vs. V_0_, and the Lineweaver–Burk graph are shown in Figure 2c and Figure 2d, respectively. Data were fitted to the Michaelis–Menten model, and the kinetic parameters are shown in Table 1.

Our kinetic data showed that the K_M_ of ASNase increased 1.66-fold when encapsulated. This means ASNase-BMV nanobioreactors had a lower affinity to Asn. The catalytic constant (K_cat_) and the K_cat_/K_M_ ratio decreased 2.17- and 3.6-fold, respectively, which means that the enzyme is less efficient when encapsulated. We expected these results because an encapsulated enzyme has diminished freedom of movement to perform the catalysis, as has been pointed out by different authors [31,35,38,42,43,44]. However, when comparing our results to the work of Díaz-Barriga et al. [38], we observed that the effect of the confinement in the K_M_ value is lower when ASNase is encapsulated through electrostatic interactions instead of when a covalent attachment is used.

In summary, the catalytic efficiency of ASNase decreased when encapsulated into BMV capsids. However, the negative impact on the kinetic parameters is lower when the enzyme is encapsulated through electrostatic interactions into BMV instead of through a covalent attachment using P22 VLPs.

### 3.3. Thermal Stability of Asparaginase into BMV Nanobioreactors

It has been proposed that enzymes could improve their stability when encapsulated into VLPs and other inorganic nanocarriers, resulting in improved medical formulations [45,46,47]. The effect of the encapsulation of ASNase into BMV-VLPs as a function of the temperature was analyzed. ASNase and ASNase-BMV were incubated at 25 °C and 37 °C for 192 h, and thermal efficiency was measured, as shown in Figure 3a and Figure 3b, respectively. As normalization criteria, enzymatic activity under no thermal stress was used as a control.

According to our results, both at 25 °C and 37 °C, ASNase showed a lower relative enzymatic efficiency as the temperature and exposure time increased. Contrary to our intuition, ASNase thermal stability decreased when encapsulated into BMV-VLPs. A similar observation was reported by Díaz-Barriga et al. [38] for ASNase-P22 nanobioreactors; however, in their system, Díaz-Barriga observed that the enzyme became even more labile when encapsulated into P22-VLPs rather than BMV-VLPs. Comparing our results, ASNase-P22 and ASNase-BMV reduced their enzymatic activity by 75% and 20%, respectively, when incubated at 25 °C for 60 h, whereas ASNase-P22 and ASNase-BMV showed reduced enzymatic activity in 50% and 32% when incubated at 37 °C for 24 h. To rule out nanobioreactors collapsed after thermal exposure, we analyzed their size by DLS, finding no alterations.

In summary, ASNase encapsulation into BMV-VLPs reduced enzymatic thermal stability, although this negative effect was lower in comparison when P22-VLPs were used. This is the first report that the type of VLP used as a nanocontainer and/or the encapsulation strategy used influences the enzyme thermal stability. Further research should be performed to enhance nanobioreactors’ stability and to elucidate the role of the molecular structure of VLPs used as nanocontainers in enzymatic thermal stability.

### 3.4. Cytotoxicity Evaluation of ASNase-BMV Nanobioreactors

To perform the cytotoxicity analysis, the T-cell leukemic line, MOLT-4, and the B-cell leukemic Reh cell line were used. MOLT-4 cell line cultures have been widely used in the literature to investigate metabolic changes following ASNase treatment; meanwhile, Reh cells were chosen for comparison purposes [48,49]. Cells were exposed to 1 IU/mL of free ASNase or BMV-ASNase for 132 h. To determine the cytotoxic effect of ASNase-BMV, we used: (a) untreated cells, (b) cells incubated with assembly buffer, (c) cells incubated with BMV-VLPs, and (d) cells incubated with free ASNase (Leunase, commercial asparaginase) as controls. Cytotoxicity results of MOLT-4 and Reh cells are shown in Figure 4a and Figure 4b, respectively.

According to our results, neither empty BMV VLPs nor the assembly buffer had cytotoxic effects over both MOLT-4 and Reh cell lines used in this study. In contrast, commercial ASNase and ASNase-BMV showed equivalent cytotoxicity against MOLT-4 and Reh cells. We emphasize that MOLT-4 cells were more sensitive both to ASNase and ASNase-BMV than Reh cells, in agreement with the literature. The half-minimal inhibitory concentration (IC_50_) of MOLT-4 and Reh was 1.2 UI/mL and 3.5 UI/mL, respectively [50,51]. Previously, Díaz-Barriga et al. [38] reported that ASNase-P22 showed a three-fold lower cytotoxicity than free ASNase. In contrast, ASNase-BMV showed similar cytotoxicity to commercial ASNase. These results can be attributed to the molecular movement of the encapsulated enzyme when electrostatic interactions are used for encapsulation, unlike covalent attachments, favoring the catalytic activity.

Taken together, our results showed that ASNase-BMV could be a better therapeutic alternative than ASNase-P22. Unlike ASNase-P22, ASNase-BMV showed similar cytotoxicity compared to commercial ASNase.

### 3.5. Humoral Responses in Mice Immunized with ASNase-BMV Nanobioreactors

In the next step, we assessed the humoral immune response induced by BMV capsids, and also described the effect of the release and stability on the ASNase activity using ASNase-BMV nanobioreactors. BALB/c female mice were vaccinated intraperitoneally with 90 μg of either empty, ASNase-filled BMV-VLPs, or ASNase on day 0 and boosted twice more on days 14 and 28. Serum samples were collected on day 0 before the first immunization and subsequently on days 7, 21, 35, and 42. Total specific IgG responses were determined against BMV-VLPs and ASNase by ELISA. Specific end-point antibody titers were calculated as the reciprocal of serum dilution where 50% of the response was observed. Our results showed specific IgG titers over time against BMV-VLPs in groups immunized with empty capsids or filled capsids, respectively (Table 2).

Anti-BMV-VLPs serum IgG was present since the 7th dpi and their titers were increased until the 42nd day (Figure 5), approximately 58 and 115-fold more in groups A and B, respectively. Filled ASNase-BMV-VLPs were more potent at inducing IgG antibodies than empty BMV-VLPs due to different protein capsid content (300 vs. 90 μg, respectively). This could be also explained by differences in the aggregation process that we observed on the BVM-VLPs over time, inducing changes in the pattern recognition molecules (PRM) affecting the humoral response. Although Nuñez-Rivera, et al. showed nonactivation of macrophages using BMV-VLPs, in vivo studies demonstrated a high IgG production, and it would be interesting to characterize IgG subtype and cytokine patterns to understand part of the immunogenic mechanism of BMV nanocarriers. No other reports have shown this humoral immune analysis. Recently, Zinkhan and collaborators [52] described the importance of the shape and size of CCMV-VLPs for enhancing IgG titers in murine models, their capacity to boost antibody production, and their feasibility as nanobioreactors.

To propose BMV-VLPs nanocarriers as a platform to have a managed release of different cargo molecules, we encapsulated ASNase and characterized specific production of anti-ASNase antibodies. Surprisingly, ASNase-BMV was more efficient at inducing anti-ASNase antibodies than free ASNase (Table 3 and Figure 5), but no statistical difference was observed. However, we detected a 47-fold increase in IgG titer from day 7 to 42 using encapsulated ASNase, while there was only a 32-fold increase when using free ASNase. Comparing the production of anti-ASNase antibodies on day 42, it was 5.7-fold higher using encapsulated ASNase, demonstrating its bioavailability, its stability, and avoiding its degradation.

This is a discouraging result for the use of BMV capsids as potential nanocarriers for the stabilization of enzymes such as asparaginase, since they showed high immunogenic reactivity, which is detrimental to the study’s objectives of reducing immunogenicity and, therefore, increasing bioavailability of asparaginase to increase treatment time. However, it opens the possibility of using these types of nanocontainers as carriers of specific antigens to increase the immune response, for example, for the generation of vaccines. In that sense, Shukla, et al. [53] demonstrated the versatility and efficacy of plant viruses’ VLPs as in situ vaccines with encouraging results.

## 4. Conclusions

The use of VLPs from plant viruses offers broad advantages for biomedical applications. Plant virus nanoparticles are not infectious toward mammalian cells and have been proposed as the next generation of nanovehicles for therapeutic delivery. We found that ASNase-BMV nanobioreactors showed a better performance than ASNase-P22. These results can be attributed to the fact that encapsulation of ASNase-BMV was accomplished through electrostatic interactions instead of covalent attachment as in ASNase-P22. In vivo studies showed the potential of BMV-VLPs as a tool to contain active enzymes and different cargo molecules, mainly those used in cancer treatment assuring their stability and versatility in finding new pharmaceuticals for treatment. To our knowledge, this is the first work where ASNase is encapsulated into plant-derived virus-like particles. In addition, this is a pioneering work where ASNase has been tested for the first time in an animal model when encapsulated into VLPs. Despite the fact that we detected specific antibodies against both BMV and ASNase over time, further research is encouraged to explore the type of immune response triggered by ASNase-BMV nanobioreactors in order to reduce immunogenicity and to improve enzymatic activity in animal models, as well as detailed pharmacodynamic studies in vivo.

## Figures and Tables

**Figure 1 pharmaceutics-15-02260-f001:**
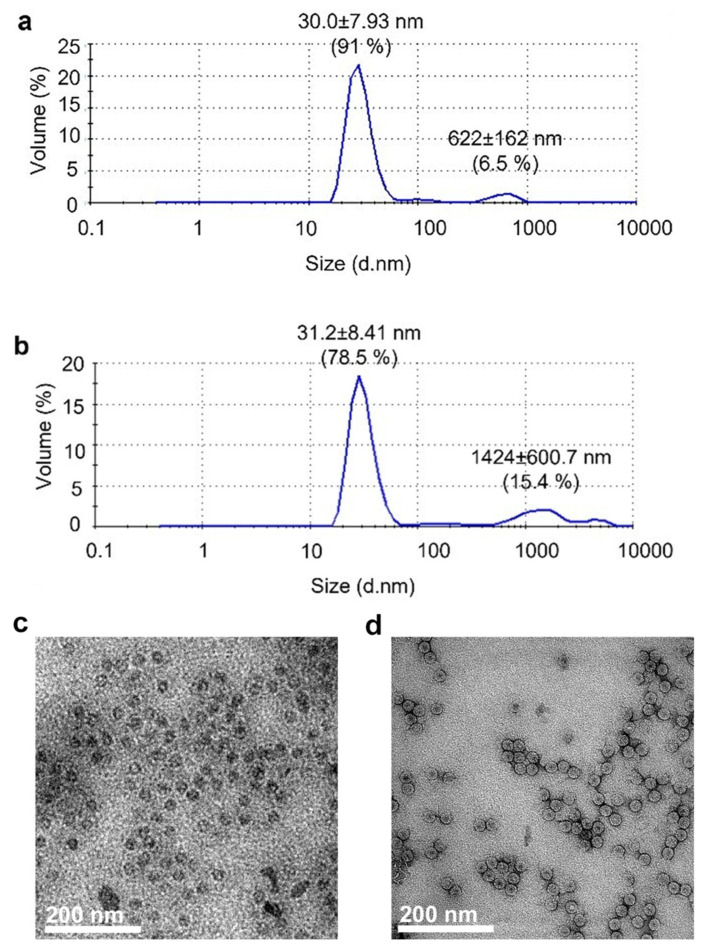
Size and morphology of BMV-VLP and ASNase-BMV nanobioreactors. Analysis by DLS of (**a**) BMV-VLP and (**b**) ASNase-BMV. TEM images of (**c**) BMV-VLP and (**d**) ASNase-BMV.

**Figure 2 pharmaceutics-15-02260-f002:**
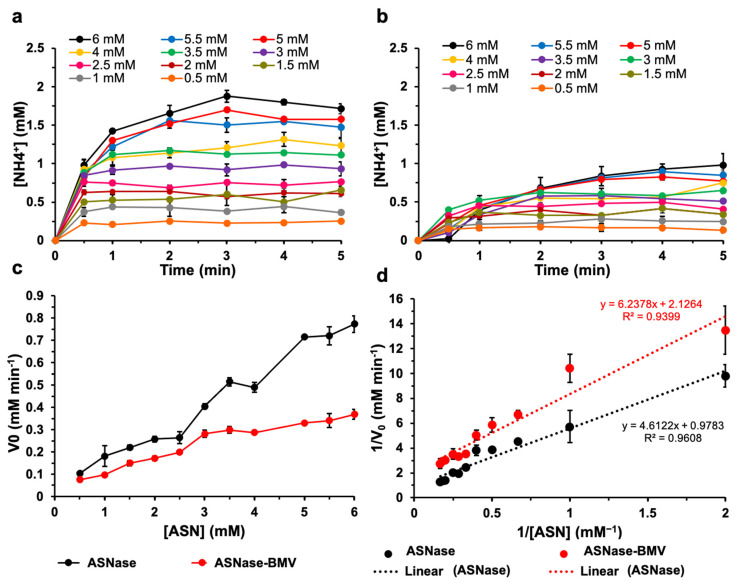
Characterization of the reaction kinetics of ASNase-BMV nanobioreactors. Enzymatic activity of (**a**) free ASNase and (**b**) ASNase-BMV nanobioreactors. Reactions were followed by measuring the ammonia production over time and varying the amount of substrate, Asn. (**c**) Graphs of the concentration of asparagine (Asn) vs. the initial velocity (V_0_). (**d**) Lineweaver–Burk graphs. Black and red lines represent the values for free ASNase and ASNase-BMV nanobioreactors, respectively. Dotted lines show the linear fit of the data, resulting equations are indicated in the figure. Pearson’s correlation coefficients (R^2^) are shown (*n* = 3). Error bars represent standard deviations.

**Figure 3 pharmaceutics-15-02260-f003:**
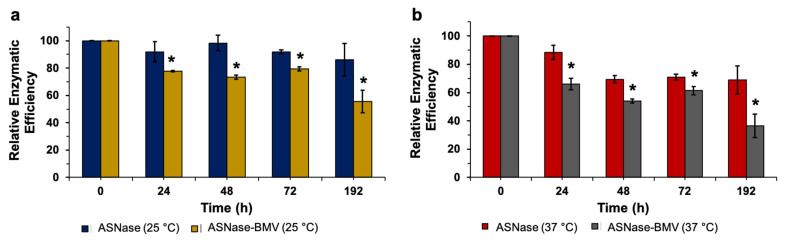
Relative enzymatic efficiency of free ASNase and ASNase-BMV nanobioreactors incubated at (**a**) 25 °C and (**b**) 37 °C. (*n* = 3). Error bars represent standard deviations. (*) *p* < 0.05.

**Figure 4 pharmaceutics-15-02260-f004:**
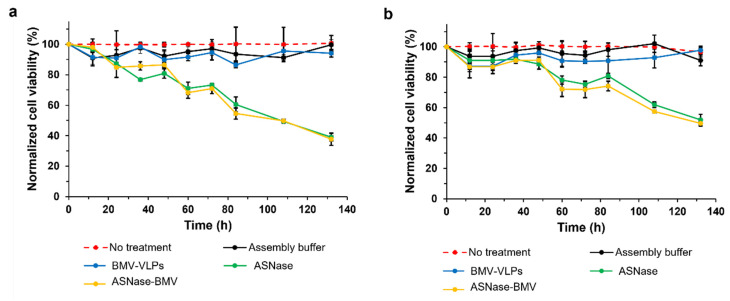
Normalized cell viability of (**a**) MOLT-4 and (**b**) Reh exposed to no treatment (red), assembly buffer (black), empty BMV-VLPs (blue), 1 Ul/mL of ASNase (green), and 1 UI/mL of ASNase-BMV nanobioreactors (yellow). Viabilities were normalized to those of untreated cells (*n* = 3). Error bars represent standard deviations.

**Figure 5 pharmaceutics-15-02260-f005:**
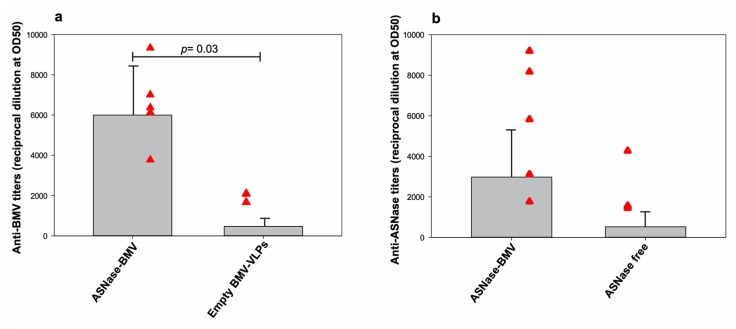
Titers of specific IgG responses on the 42nd day post first immunization. (**a**) anti-BMV specific IgG antibodies. Mice immunized with ASNase-BMV (filled capsids) and empty BMV-VLPs were analyzed. (**b**) anti-ASNase specific IgG antibodies. Mice immunized with ASNase-BMV (filled capsids) and free ASNase were analyzed. Serum samples were diluted 1:10 then, 1:3 serial dilutions were performed. OD50 specific titers were calculated as reciprocal dilution values of data. The mean and standard deviation of 5 mice per group (red triangles) were plotted in gray bars), showing the individual mouse responses for each group. Student’s *t*-tests were used to calculate the significance between groups.

**Table 1 pharmaceutics-15-02260-t001:** Kinetic parameters of free ASNase and ASNase-BMV nanobioreactors.

Enzyme	K_M_ (mM)	V_max_ (mM min^−1^)	K_cat_ × 10^11^ (min^−1^)	V_max_/K_M_ (min^−1^)	K_cat_/K_M_ × 10^10^ (mM^−1^ min^−1^)
ASNase (free)	2.83	1.02	1.56	0.36	5.51
ASNase-BMV nanobioreactors	4.71	0.47	0.72	0.10	1.53

K_M_: Michaelis–Menten constant; V_max_: maximum velocity; K_cat_: catalytic constant.

**Table 2 pharmaceutics-15-02260-t002:** End-point titer of specific anti-BMV-VLPs IgG.

	Days Post First Immunization				
Mice Groups	Description	7th	21st	35th	42nd
**A**	Anti-BMV-VLPs (empty capsids)	8.1 ± 6.0	204.4 ± 107.0	418.8 ± 201.2	470.6 ± 275.9
**B**	Anti-BMV (ASN-BMV)	52.3 ± 43.6	3471.9 ± 3355.4	5449.3 ± 3024.5	5995.5 ± 2348.9

Values represent mean antibody titer of five mice ± standard deviation on 7th, 21st, 35th and 42nd days post first immunization (dpi). Group A was immunized with empty BMV-VLPs and group B was immunized with ASNase encapsulated into BMV-VLPs. *p* = 0.03.

**Table 3 pharmaceutics-15-02260-t003:** The end-point titer of specific anti-ASNase IgG. Values represent mean antibody titer of five mice ± standard deviation on 7th, 21st, 35th, and 42nd days post first immunization (dpi). Group B was immunized with 90 µg of ASNase encapsulated into BMV-VLPs and group D was immunized with 90 µg of free ASNase (Leunase, Sanfer).

	Days Post First Immunization				
Mice Groups	Description	7th	21st	35th	42th
**B**	Anti-ASNase (ASN-BMV)	63.5 ± 43.5	727.1 ± 419.7	1529.1 ± 1210.7	2973.6 ± 2195.7
**D**	Anti-ASNase (free ASN)	16.5 ± 18.5	125.6 ± 216.5	492.8 ± 928	521.8 ± 852.4

## Data Availability

The data presented in this study are available in this article.

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
