# Peer review of "A Novel Formulation of Asparaginase Encapsulated into Virus-like Particles of Brome Mosaic Virus: In Vitro and In Vivo Evidence"

_pharmaceutics, 2023, doi:10.3390/pharmaceutics15092260_

Round 1

Reviewer 1 Report

The paper entitled  A Novel Formulation of Asparaginase Encapsulated into Virus-1 like Particles of Brome Mosaic Virus: In Vitro and In Vivo Evidence” by Francisca Villanueva-Flores, Ana Ruth Pastor and Laura A. Palomares, and Alejandro Huerta-Saquero deals with very ambitious topic concerning production of nanobireactors composed asparaginase used in treatment of Acute Lymphoblastic Leukemia (ALL) in children encapsulated in Brome Mosaic Virus-derived VLP. This topic is of high interest to virology, nanotechnology and medicine.

Unfortunately, the presented results and conclusions are unclear, and the work requires deep corrections. Below are the issues that need clarification:

1.       Line 68 – BMV is icosahedral, according to most of scientific literature.

2.       Line 76-80. The part concerning pI is unclear. The pI of BMV CP alone is about 9. The pI of VLP depends on what is packaged inside. Usually the native virions with pI about 5 are not delivered to the cells, so this part should be clarified.

3.       Line 155. The Authors use MWCO 6000-8000 membrane to remove the non-encapsulated enzyme. However, the enzyme is big tetrameric complex with molecular weight much higher than 6000-8000 Daltons and it is impossible to remove it from the solution with such membrane.

4.       Taking the above to account, the results on enzymatic activity and other biological properties ASN-VLPs are not convincing.

5.       The description of experiments on mice immunization is unclear. “Group A was immunized with 90 μg of empty BMV-VLPs, group B was immunized with 90 μg of ASN encapsulated into BMV-VLPs, Group C, as a control group, was immunized with formulation buffer; and group D was immunized with 90 μg of commercial ASN”. It is unclear, how the group B was immunized. Either the animals were immunized with 90ug of total of ASN enclosed in BMV-VLPs (with ratio ASN/BMV = 0.07) or 90ug ASN enclosed in some amount of VLPs the groups A, B and C seem to be not comparable.

6.       Figure 1. The TEM images of empty BMV VLPs and ASN BMV VLPs should be presented in the same size scale. The method of assessment of packaged versus empty VLPs, relying on arbitrary decision is not convincing.

7.       Citation no. 37 should be corrected.

Author Response

The paper entitled  “A Novel Formulation of Asparaginase Encapsulated into Virus-1 like Particles of Brome Mosaic Virus: In Vitro and In Vivo Evidence” by Francisca Villanueva-Flores, Ana Ruth Pastor and Laura A. Palomares, and Alejandro Huerta-Saquero deals with very ambitious topic concerning production of nanobireactors composed asparaginase used in treatment of Acute Lymphoblastic Leukemia (ALL) in children encapsulated in Brome Mosaic Virus-derived VLP. This topic is of high interest to virology, nanotechnology and medicine.

RESPONSE: Thank you very much for your comments! We appreciate your time and effort to make this a better contribution.

Unfortunately, the presented results and conclusions are unclear, and the work requires deep corrections. Below are the issues that need clarification:

  1. Line 68 – BMV is icosahedral, according to most of scientific literature.

RESPONSE: Corrected.

  1. Line 76-80. The part concerning pI is unclear. The pI of BMV CP alone is about 9. The pI of VLP depends on what is packaged inside. Usually the native virions with pI about 5 are not delivered to the cells, so this part should be clarified.

RESPONSE: According to Duran-Meza, et. al. (2021) the isoelectric point (pI) of native BMV is 5.2, whereas pI of BMV CP is 6.5 and seems to be independent of the cargo molecules (Duran-Meza AL, Villagrana-Escareño MV, Ruiz-García J, Knobler CM, Gelbart WM. Controlling the surface charge of simple viruses. PLoS One 2021;16(9):e0255820).

On the other hand, for ALL treatment, it is desirable that nanoreactors not enter the cell but degrade circulating asparagine from the bloodstream.

We modify the paragraph for clarification as follows:

“The isoelectric point (pI) of native BMV is 5.2, and the pI of BMV CP is 6.5. At the pI, electrical charges are fully neutralized, and hydrophobic interactions dominate and stabilize the capsid. This property is advantageous considering that acid pH is typical of cancerous microenvironments because of anaerobic glycolysis [25], allowing the use of enzymatic nanoreactors for cancer treatment.” 

  1. Line 155. The Authors use MWCO 6000-8000 membrane to remove the non-encapsulated enzyme. However, the enzyme is big tetrameric complex with molecular weight much higher than 6000-8000 Daltons and it is impossible to remove it from the solution with such membrane.

RESPONSE: You are right. Thank you. We omitted to describe the complete purification procedure. The paragraph was changed as follows:

“To remove the non-encapsulated enzyme, ASNase-BMV nanobioreactors were ultracentrifuged at 32,000 rpm for 2 h at 4 °C on a 10% sucrose cushion. Pellets were resuspended with 500 µL of PBS and then dialyzed with the same buffer at 4 °C for 24 h using Spectra/Por® 1 MWCO 6000-8000 membranes (Spectrum® Laboratories)”.

  1. Taking the above to account, the results on enzymatic activity and other biological properties ASN-VLPs are not convincing.

RESPONSE: Agree. Purification of nanoreactors is essential to characterize biological and enzymatic properties. By ultracentrifugation on sucrose cushion, the non-encapsulated enzyme remains in the supernatant, whereas encapsulated asparaginase inside BMV capsids precipitates, allowing their purification. Indeed, we observed different kinetic parameters in encapsulated asparaginase compared to the free enzyme, confirming that capsids were purified from free-asparaginase.

We added this information in discussion section.

  1. The description of experiments on mice immunization is unclear. “Group A was immunized with 90 μg of empty BMV-VLPs, group B was immunized with 90 μg of ASN encapsulated into BMV-VLPs, Group C, as a control group, was immunized with formulation buffer; and group D was immunized with 90 μg of commercial ASN”. It is unclear, how the group B was immunized. Either the animals were immunized with 90ug of total of ASN enclosed in BMV-VLPs (with ratio ASN/BMV = 0.07) or 90ug ASN enclosed in some amount of VLPs the groups A, B and C seem to be not comparable.

RESPONSE: Agree. Thank you very much for pointing out this methodological mistake. Samples were adjusted to administer the same amount of asparaginase, encapsulated or free. In that sense, we administered a lower amount of BMV-VLPs protein (90 µg) than we should to be comparable with the amount of capsids protein administered with nanoreactors with asparaginase inside (which amount corresponds to ca. 300 µg of BMV-VLPs containing 90 µg of asparaginase).

The immunization protocol was corrected as follows:

“Group A was immunized with 90 µg of empty BMV-VLPs; group B was immunized with 90 µg of ASN encapsulated into BMV-VLPs (containing around 300 µg of BMV-VLPs protein); Group C, as a control group, was immunized with formulation buffer; and group D was immunized with 90 µg of commercial asparaginase (Leunase).”

As you stated, our results are not fully comparable between Group A (90 µg of BMV-VLPs) vs. Group B (300 µg of BMV-VLPs + 90 µg asparaginase), in which we obtained a higher immune response against BMV-VLPs in Group B than Group A, accordingly to VLPs amount administered. However, we can establish a valid comparison between free asparaginase vs. encapsulated asparaginase, the experiment's primary purpose.

The discussion of the results was corrected accordingly.

  1. Figure 1. The TEM images of empty BMV VLPs and ASN BMV VLPs should be presented in the same size scale. The method of assessment of packaged versus empty VLPs, relying on arbitrary decision is not convincing.

RESPONSE: Figure 1 was replaced with TEM images at the same size scales. Visual differences between packaged and empty capsids are unclear, so this information was omitted.

  1. Citation no. 37 should be corrected.

RESPONSE: Done.

Reviewer 2 Report

This study describing using a Brome Mosaic Virus-like (BMV-VLPs) particles to encapsulate asparaginase (ASNase) which shows promising results including

01.  Equivalent cytotoxicity of empty vehicle BMV-VLPs to untreated and encapsulate ASNase in BMV (ASNase-BMV) to naked ASNase,

02.  Better reaction kinetics and enzymatic efficiency (enzymatic thermal stability) of ASNase-BMV than ASNase-p22,

03.  Better anti-ASN antibodies induction than free ASNase,

04.  but higher immunogenic reactivity.

This study was well designed and carried out.  Although more characterization could be carried out, e.g., loading efficiency, loading capacity, drug release, storage stability (e.g., Rh measurement as a function of time, aggregation measurement using ulcentrifugation, etc.), I do not think it is necessary as this study is intended to explore the option of using BMV-VLPs and alike particles to encapsulate ASNase for future therapeutic use.  However, I will expect more characterization work to be done when a final promising BMV-VLP like nano carrier is found and selected to be used for clinical trials, and I would like to review the manuscript when authors reach to this stage.

These are several comments I have for authors and I would recommend authors to revise according to my comments.

01.  In all figures, I would recommend using the same color for both data points and error bars for a single data series, as all series have black error bars and it is difficult to distinguish statistical differences.

02.  In Figure 4, ASNase-BMV shows similar cytotoxicity as the free ASNase.  Is it due to substantial ASNase release immediately after treatment?  What are 1/2 life of free ASNase and ASNase-BMV?  If 1/2 life are different, then what other factors could contribute to the same cytotoxicity?  Please provide some explanations and insights.

03.  In Figure 5 and Table 3, I find better anti-ASNase antibody induction and higher immunogenicity are contradictory yet not mutually exclusive. Would fast ASNase release cause this phenomenon?  Please explain.

I would need authors to revise their manuscript according to my comments and I would recommend publishing this manuscript as it would be of interest to readers in this field.

Author Response

This study describing using a Brome Mosaic Virus-like (BMV-VLPs) particles to encapsulate asparaginase (ASNase) which shows promising results including

  1.  Equivalent cytotoxicity of empty vehicle BMV-VLPs to untreated and encapsulate ASNase in BMV (ASNase-BMV) to naked ASNase,
  2.  Better reaction kinetics and enzymatic efficiency (enzymatic thermal stability) of ASNase-BMV than ASNase-p22,
  3.  Better anti-ASN antibodies induction than free ASNase,
  4.  but higher immunogenic reactivity.

This study was well designed and carried out.  Although more characterization could be carried out, e.g., loading efficiency, loading capacity, drug release, storage stability (e.g., Rh measurement as a function of time, aggregation measurement using ulcentrifugation, etc.), I do not think it is necessary as this study is intended to explore the option of using BMV-VLPs and alike particles to encapsulate ASNase for future therapeutic use.  However, I will expect more characterization work to be done when a final promising BMV-VLP like nano carrier is found and selected to be used for clinical trials, and I would like to review the manuscript when authors reach to this stage.

RESPONSE: Thank you very much for your comments! We appreciate your time and effort to make this a better contribution.

We agree. A complete physicochemical and pharmacological characterization will be necessary for the next steps, such as clinical trials. Unfortunately, we found high immunogenic activity of BMV capsids and encapsulated asparaginase, which is detrimental to the objective of using BMV as an asparaginase carrier for ALL treatment.

Nevertheless, to our knowledge, we want to point out that this is the first report of encapsulation of asparaginase in VLP´s from BMV and the immunogenic test of nanoreactors in mice.

These are several comments I have for authors and I would recommend authors to revise according to my comments.

  1.  In all figures, I would recommend using the same color for both data points and error bars for a single data series, as all series have black error bars and it is difficult to distinguish statistical differences.

RESPONSE: Thank you, but we do not consider modifying error bars necessary since asterisks are used to point out statistical differences between groups when necessary.

  1.  In Figure 4, ASNase-BMV shows similar cytotoxicity as the free ASNase.  Is it due to substantial ASNase release immediately after treatment?  What are 1/2 life of free ASNase and ASNase-BMV?  If 1/2 life are different, then what other factors could contribute to the same cytotoxicity?  Please provide some explanations and insights.

RESPONSE: MOLT-4 cell viability assays were chosen to demonstrate the functionality of asparaginase encapsulated into BMV VLPs. Encapsulated or free asparaginase toxicity depends on the enzymatic activity degrading asparagine from culture media, provoking cell arrest and death. The release of asparaginase from the nanoreactor is not necessary for obtaining enzymatic activity; on the contrary, it is convenient for our purposes to maintain asparaginase confined into nanoreactors to protect it from proteases and immune response.

The half-life of free asparaginase and ASNase-BMV was not determined in cell cultures, but we can observe that after 130 hr, both asparaginase formulations remain active.

  1.  In Figure 5 and Table 3, I find better anti-ASNase antibody induction and higher immunogenicity are contradictory yet not mutually exclusive. Would fast ASNase release cause this phenomenon?  Please explain.

RESPONSE: The higher rise of antibodies from encapsulated asparaginase was unexpected and undesirable for using BMV VLPs as carriers for asparaginase. Nevertheless, due to their proteic nature, this phenomenon can be explained by the immunogenic response raised by BMV capsids alone.

The protein capsids from the ASNase-BMV nanoreactor seem to increase immune response by acting as an adjuvant, concomitantly increasing anti-asparaginase antibody production and immunogenicity.

We expand the discussion section abording this matter.

I would need authors to revise their manuscript according to my comments and I would recommend publishing this manuscript as it would be of interest to readers in this field.

RESPONSE: Thank you.

Reviewer 3 Report

The research presents a very complex approach to developing and characterizing ASNase-BMV nanoparticles for therapeutic purposes. I find the results interesting and suitable for publishing in Pharmaceutics, but not before a revision. My main concerns are regarding the presentation and structure of the manuscript. Here are some recommendations:

Materials and methods: 

-        Line 200 – What is the concentration of BMV-VLPs as a coating antigen? 

-       Lines 206 & 213 – “developed” might be substituted with a more suitable word.

-   Lines 207 & 215 – It is not clear to me. Better to be precise in the description of the ELISA development in general. 

-       Line 209 – “home-made” might be substituted with “in-house”.

-       Line 217 – “expressed” might be substituted with “displayed”.

Results and Discussion:

-          Split it into 2 sections “Results” and “Discussion”.

-       Figures and Tables must be displayed after the first mention in the text (e.g. Figure 1 and Table 1).

-    Table 2 and Table 3: Specify the source of the given numbers in the description or on the table itself.

-       Figure 5: It is better to present the two sections with the same scale on the y-axis. It may lead to false conclusions in the current state.

-         Line 336-338 – The sentence is too complex and hard to read.

Other:

-   The manuscript must be structured and formatted according to the journal’s criteria.

-     Ethical statement for the animal experiment must be added.

Author Response

The research presents a very complex approach to developing and characterizing ASNase-BMV nanoparticles for therapeutic purposes. I find the results interesting and suitable for publishing in Pharmaceutics, but not before a revision. My main concerns are regarding the presentation and structure of the manuscript. Here are some recommendations:

RESPONSE: Thank you very much for your comments! We appreciate your time and effort to make this a better contribution.

Materials and methods: 

-        Line 200 – What is the concentration of BMV-VLPs as a coating antigen? 

RESPONSE: According to asparaginase-BMV capsid protein ratio, the amount of BMV-VLPs used was 300 µg. For clarification, the immunization protocol was modified as follows:

“Group A was immunized with 90 µg of empty BMV-VLPs; group B was immunized with 90 µg of ASN encapsulated into BMV-VLPs (containing around 300 µg of BMV-VLPs protein); Group C, as a control group, was immunized with formulation buffer; and group D was immunized with 90 µg of commercial ASN.”

-       Lines 206 & 213 – “developed” might be substituted with a more suitable word.

RESPONSE: Corrected.

-   Lines 207 & 215 – It is not clear to me. Better to be precise in the description of the ELISA development in general. 

RESPONSE: We re-write ELISA description as follows:

“Anti-BMV IgG antibodies were measured in mice sera samples by ELISA. 100 µL of a solution of purified BMV-VLP at 5µg/mL were adsorbed onto 96-well microplates for at least 16 hours at 2-8 °C. Following the incubation time, plates were washed three times with Tris buffer to remove unbound BMV-VLP and blocked with 2 mg/mL of gelatin in Tris buffer pH 8 to prevent non-specific binding and incubated 2 h at 37 °C. Then, serial dilutions of mice sera at 100 µL of final volume were prepared and loaded into the plate. After 2 h of incubation at 37 °C, plates were washed five times, and an alkaline phosphatase-conjugated goat anti-mouse IgG antibody (Jackson ImmunoResearch, USA) in a 1:1500 dilution was added. Plates were incubated 1 h at 37 °C. Next, plates were washed again five times, 100 µL of phosphatase substrate at 120 µg/mL (Sigma, P4744) was added and read at 405 nm in a microplate reader (FLUOstar Omega, BMG Labtech, EE. UU). IgG titers were calculated as the reciprocal of the serum dilution at which the response was 50%.

Anti-ASNase IgG antibodies were measured with an in-house ELISA assay. Briefly, 100 µL of a solution of ASNase at 5 µg/mL was adsorbed into 96-well plates. After at least 16 h of incubation at 2-8 °C, plates were washed and blocked for 2 h at 37 °C with a 2 mg/mL gelatin solution. 1:3 serial dilutions of mice sera were prepared and added to plates and incubated for 2 h at 37 °C, then plates were washed, and an alkaline phosphatase-conjugated goat anti-mouse IgG antibody (Jackson ImmunoResearch, USA) in a 1:1500 dilution was added. One hour later, plates were added with 100 µL of phosphatase substrate at 120 µg/mL (Sigma, P4744) and read at 405 nm in a microplate reader. Titers were calculated as the reciprocal of the serum dilution at which the response was 50%”.

-       Line 209 – “home-made” might be substituted with “in-house”.

RESPONSE: Corrected.

-       Line 217 – “expressed” might be substituted with “displayed”.

RESPONSE: Corrected. 

Results and Discussion:

-          Split it into 2 sections “Results” and “Discussion”.

RESPONSE: Thank you for the suggestion. For clarity purposes, we consider that the paper presented in the current form is acceptable.

-       Figures and Tables must be displayed after the first mention in the text (e.g. Figure 1 and Table 1).

RESPONSE: Corrected.

-    Table 2 and Table 3: Specify the source of the given numbers in the description or on the table itself.

RESPONSE:

Detailed description was done in the table titles, results and discussion section as follow:

Total specific IgG responses were determined against BMV-VLPs and ASNase by ELISA. Specific end-point antibodies titers were calculated as reciprocal of serum dilution where 50% of response was observed. Our results showed specific IgG titers over time against BMV-VLPs in groups immunized with empty capsids or filled capsids, respectively. (Table 2).

Table 2. End-point titer of specific anti-BMV-VLPs IgG. Values represent mean antibody titer of five mice ± standard deviation on 7th, 21st, 35th and 42nd days post first immunization (dpi). Group A was immunized with empty BMV-VLPs and group B was immunized with ASNase encapsulated into BMV-VLPs. p=0.03

Table 3. The end-point titer of specific anti-ASNase IgG. Values represent mean antibody titer of five mice ± standard deviation on 7th, 21st, 35th, and 42nd days post-first immunization (dpi). Group B was immunized with 90 µg of ASNase encapsulated into BMV-VLPs and group D, was immunized with 90 µg of free ASNase (Leunase, Sanfer).

-       Figure 5: It is better to present the two sections with the same scale on the y-axis. It may lead to false conclusions in the current state.

RESPONSE: Corrected

-         Line 336-338 – The sentence is too complex and hard to read.

RESPONSE: Corrected.

Other:

-   The manuscript must be structured and formatted according to the journal’s criteria.

RESPONSE: Thank you. The paper will be formatted accordingly.

-     Ethical statement for the animal experiment must be added.

RESPONSE: Thank you. The bioethics committee approval letter was sent to editorial office.

We added this information to the paper:

“The IBt-Bioethics Committee on the Use and Care of Animals approved all experiments.”